# Confidence Sets for Statistical Classification

**Wei Liu [1],\*, Frank Bretz [2],† , Natchalee Srimaneekarn [1], Jianan Peng [3] and Anthony J. Hayter [4]**

[1]  S3RI and School of Mathematics, University of Southampton, Highfield, Southampton SO17 1BJ, UK
[2]  Novartis Pharma AG, 4002 Basel, Switzerland
[3]  Department of Mathematics and Statistics, Acadia University, Wolfville, NS B4P 2R6, Canada
[4]  Department of Statistics and Operations Technology, University of Denver, Denver, CO 80208-8921, USA
\*  Correspondence: w.liu@maths.soton.ac.uk
†  Visiting address: Center for Medical Statistics, Informatics and Intelligent Systems,
   Medical University of Vienna, Vienna 1090, Austria.

**Abstract:** Classification has applications in a wide range of fields including medicine, engineering, computer science and social sciences among others. In statistical terms, classification is inference about the unknown parameters, i.e., the true classes of future objects. Hence, various standard statistical approaches can be used, such as point estimators, confidence sets and decision theoretic approaches. For example, a classifier that classifies a future object as belonging to only one of several known classes is a point estimator. The purpose of this paper is to propose a confidence-set-based classifier that classifies a future object into a single class only when there is enough evidence to warrant this, and into several classes otherwise. By allowing classification of an object into possibly more than one class, this classifier guarantees a pre-specified proportion of correct classification among all future objects. An example is provided to illustrate the method, and a simulation study is included to highlight the desirable feature of the method.

**Keywords:** classification; confidence level; confidence set; coverage frequency; simultaneous tolerance intervals; statistical inference

---

## 1. Introduction

Classification has applications in a wide range of fields including medicine, engineering, computer science and social sciences, among others. See, e.g., the recent books by [1–4]. Classical examples include medical diagnosis, automatic character recognition, data mining (such as credit scoring, consumer sales analysis and credit card transaction analysis) and artificial intelligence (such as the development of machines with brain-like performance). As many important developments in this area are not confined to the statistics literature, various other names, such as supervised learning, pattern recognition and machine learning, have been used. In recent years, there have been many exciting new developments in both methodology and applications, taking advantage of increased computational power readily available nowadays. Broadly speaking, classification methods can be divided into probabilistic methods (including Bayesian classifiers), regression methods (including logistic regression and regression trees), geometric methods (including support vector machines), and ensemble methods (combining classifiers for improved robustness).

A classifier is a decision rule built from a training data set $\mathcal{T}$ that classifies all future objects as belonging to one or several of the $k$ known classes, where $k$ is a pre-specified number. The drawback of a classifier that classifies each future object into only one of $k$ classes is that, when the object is close to the classification boundaries of several classes, $M(\geq 2)$ say, the chance of misclassification is close to $(M-1)/M$, which may be close to one when $M$ is large. A sensible approach in this situation is to acknowledge that such an object has similar chances of belonging to $M$ classes and hence to avoid

classifying it into only one of the *M* classes. In medical diagnosis, for example, if there is not enough evidence to classify a patient as having a disease or not, then it is wise not to give a diagnosis that is quite likely to be wrong.

Various procedures have been proposed in the literature to deal with this difficulty. One type of procedure allows a rejection option, that is, if a future object falls into a 'rejection' region, then no classification is made for the object. Such a procedure aims to construct a suitable rejection region to minimize a pre-specified risk; see, e.g., [5–9] and the references therein. Non-deterministic classifiers are proposed in [10], which allow a future object to be classified possibly into several classes. Again, such a classifier is constructed to minimize a pre-specified risk.

For the binary classification problem (i.e., $k = 2$), ref. [11] proposes to find two 'tolerance' regions (corresponding to the two classes) in the feature/predictor space, with a specific coverage level for each class that minimize the probability that an object falls into the intersection of the two tolerance regions since an object in this intersection will not be classified. This approach is akin to the decision-theoretic approaches mentioned in the last paragraph but uses this specific probability as the risk to minimize. As with other decision-theoretic approaches, it is not constructed to guarantee the proportion of correction classification and thus is different from the approach proposed in this paper. Further development of this approach is considered in [12].

The conformal prediction approach of [13,14] also classifies a future object into possibly several classes that contain the true class with a pre-specified probability. However, this approach is designed for the 'online' setting in which the true classes of all the observed objects are revealed and hence known before the classification of the next object is made. This online setting is different from the usual setting of classification considered in this paper, in which a classifier is built from the available training data set $\mathcal{T}$ and then used to classify a large number of future objects without knowing their true classes.

For the binary classification problem, ref. [15] proposes a classifier that allows no classification of an object. By controlling the size of the non-classification region (for which classification error does not occur) via a tuning constant, a 'generalized error' of the classifier is controlled at a pre-specified level with a specified confidence about the randomness in the training data set $\mathcal{T}$. The construction of this classifier is related to the tolerance sets going back to [16]. Note, however, that the algorithm in [15] may result in a different classifier if a different observation in the training data set is used as the 'base' instance in the algorithm, which is quite odd from a statistical point of view. In addition, the 'generalized error' is different from the long run frequency of correct classification, which the procedure proposed in this paper aims to control.

The purpose of this paper is to propose a classifier that classifies a future object into a single class only when there is enough evidence to warrant this, and into several classes otherwise. By allowing classification of an object into potentially more than one class, this classifier guarantees a pre-specified proportion of correct classification among all future objects. Specifically, classification of a future object is treated as a standard problem of statistical inference about the unknown parameter *c*, the true class of the object, and the confidence set approach for *c* is adopted. In order to consider the probability of correct classification, it is necessary to assume certain probability distributions for the feature measurements from the *k* classes. In this paper the feature measurements of the *k* classes are assumed to follow multivariate normal distributions, which is widely used either directly or after some transformation (see [17,18]).

The layout of the paper is as follows: Section 2 contains some preliminaries, including the idea of [19,20] from which the new approach proposed in this paper is developed. The simple situation where the means $\boldsymbol{\mu}_i$ and covariance matrices $\Sigma_i$ of the *k* multivariate normal distributions underlying the *k* classes are assumed to be known is considered in Section 3. The more realistic situation where both $\boldsymbol{\mu}_i$ and $\Sigma_i$ are unknown parameters is studied in Section 4. Section 5 provides an illustrative example. A simulation study is given in Section 6 to highlight the major advantage of the new classifier

proposed in this paper. Section 7 provides the conclusions. Finally, some mathematical details are provided in the Appendix A.

## 2. Preliminaries

Let the $p$-dimensional data vector $\mathbf{x}_i = (x_{i1}, \ldots, x_{ip})^T$ denote the feature measurement on an object from the $i$th class, which has multivariate normal distribution $N(\boldsymbol{\mu}_i, \Sigma_i)$, $i = 1, \ldots, k$. The training data set is given by $\mathcal{T} = \{\mathbf{x}_{i1}, \ldots, \mathbf{x}_{in_i}; i = 1, \ldots, k\}$, where $\mathbf{x}_{i1}, \ldots, \mathbf{x}_{in_i}$ are i.i.d. observations from the $i$th class with distribution $N(\boldsymbol{\mu}_i, \Sigma_i)$, $i = 1, \ldots, k$. The classification problem is to make inference about $c$, the true class of a future object, based on the feature measurement $\mathbf{y} = (y_1, \ldots, y_p)^T$ observed on the object, which is only known to belong to one of the $k$ classes and so follows one of the $k$ multivariate normal distributions. In statistical terminology, $c$ is the unknown parameter of interest that takes a possible value in the simple parameter space $C = \{1, \ldots, k\}$. We emphasize that $c$ is treated as non-random in our frequentist approach.

A classifier that classifies an object with measurement $\mathbf{y}$ into one single class in $C = \{1, \ldots, k\}$ can be regarded as a point estimator of $c$. The classifier proposed in this paper provides a set $\mathcal{C}_{\mathcal{T}}(\mathbf{y}) \subseteq C$ as plausible values of $c$. Depending on $\mathbf{y}$ and the training data set $\mathcal{T}$, $\mathcal{C}_{\mathcal{T}}(\mathbf{y})$ may contain only a single value, in which case $\mathbf{y}$ is classified into one single class given by $\mathcal{C}_{\mathcal{T}}(\mathbf{y})$. When $\mathcal{C}_{\mathcal{T}}(\mathbf{y})$ contains more than one value in $C$, $\mathbf{y}$ is classified as possibly belonging to the several classes given by $\mathcal{C}_{\mathcal{T}}(\mathbf{y})$. Hence, in statistical terms, the classifier proposed in this paper uses the confidence set approach. The inherent advantage of the confidence set approach over the point estimation approach is the guaranteed $1 - \alpha$ proportion of confidence sets that contain the true classes.

The confidence set for $c$ is constructed below by inverting a family of acceptance sets for testing $H_0 : c = l$ for each $l \in C$. This method of constructing a confidence set was given by [21] and has been used and generalized to construct numerous intriguing confidence sets; see, e.g., [22–29]

Now, the key idea of [19,20] is presented very briefly, which is crucial for understanding our proposed approach to classification. Assuming that response $y$ and predictor $x$ are related by a standard linear regression model $y = \alpha_0 + \alpha_1 x + \epsilon$ and a training data set $\mathcal{T}$ on $(y, x)$ is available for estimating $\alpha_0, \alpha_1$ and the error variance $\sigma^2$, refs. [19,20] consider how to construct confidence sets for the unknown (non-random) values of the predictor $x$ corresponding to the large number of future observed values of the response $y$. As the same training data set $\mathcal{T}$ is used in the construction of all these confidence sets, the randomness in the future $y$-values and the randomness in $\mathcal{T}$ clearly play different roles and thus should be treated differently. The procedure proposed in [19,20] has a probability of at least $\gamma$, with respect to the randomness in $\mathcal{T}$ that at least $1 - \alpha$ proportion of all the confidence sets, constructed from the same $\mathcal{T}$, include the true $x$-values, where $\gamma$ and $1 - \alpha$ are pre-specfied probabilities. This idea/approach has been studied by many researchers; see, e.g., [30–36] and the references therein. One fundamental result is that the confidence sets constructed from $(\gamma, 1 - \alpha)$ simultaneous tolerance intervals do satisfy the '$\gamma$-probability-$(1 - \alpha)$'-proportion property specified above. In particular, ref. [36] points out by constructing a counter example that the confidence sets constructed from $(\gamma, 1 - \alpha)$ pointwise tolerance intervals do not guarantee the '$\gamma$-probability-$(1 - \alpha)$-proportion' property in general. A similar idea is also used in [37] to construct confidence sets for the numbers of coins in all future bags with known weights.

Since a classifier is built from the training data set $\mathcal{T}$ and then used to classify a large number of future objects in terms of confidence sets for their true classes, the future observed $y$-values play similar roles as the future observed $y$-values whilst the unknown true classes of the future objects play similar roles as the unknown true $x$-values of the future observed $y$-values, in the approach of [19,20] given in the last paragraph. Hence, it is natural to adopt the approach of [19,20] to construct confidence sets for the unknown true classes of future objects with the '$\gamma$-probability-$(1 - \alpha)$-proportion' property, that is, the probability, with respect to the randomness in $\mathcal{T}$, is at least $\gamma$ that at least $(1 - \alpha)$ proportion of all the confidence sets constructed from the same $\mathcal{T}$ do include the unknown true classes of all future objects.

### 3. Known $\mu_i$ and $\Sigma_i$

In this section, the values of $\mu_i$ and $\Sigma_i$ are assumed to be known, which helps to motivate and understand the confidence sets constructed in Section 4 for the more realistic situation where the values of $\mu_i$ and $\Sigma_i$ are unknown. Since $\mu_i$ and $\Sigma_i$ are known, no training data set $\mathcal{T}$ is required to estimate $\mu_i$ and $\Sigma_i$. Hence, the confidence sets in this section are denoted as $\mathcal{C}(\mathbf{y})$, without the subscript $\mathcal{T}$.

If $\mathbf{y}$ is from the $l$th class, then $\mathbf{y} \sim N(\mu_l, \Sigma_l)$ and so $(\mathbf{y} - \mu_l)^T\Sigma_l^{-1}(\mathbf{y} - \mu_l)$ has the chi-square distribution $\chi_p^2$ with $p$ degrees of freedom. We construct a $1 - \alpha$ confidence set for the class $c$ of the observed $\mathbf{y}$ by using [21] method of inverting a family of $1 - \alpha$ acceptance sets for testing $H_0 : c = l$ for each number $l$ in $C$. Specifically, the acceptance set for $H_0 : c = l$ is given by

$$\mathcal{A}_l = \left\{ \mathbf{y} \in R^p : (\mathbf{y} - \mu_l)^T\Sigma_l^{-1}(\mathbf{y} - \mu_l) \le \lambda \right\}, \tag{1}$$

where $\lambda = \chi_{p,1-\alpha}^2$ is the $1 - \alpha$ quantile of the $\chi_p^2$ distribution. It follows directly from Neyman's method that the confidence set is given by

$$\mathcal{C}(\mathbf{y}) = \left\{ l \in C : (\mathbf{y} - \mu_l)^T\Sigma_l^{-1}(\mathbf{y} - \mu_l) \le \lambda \right\}. \tag{2}$$

It is straightforward to show, by using the Neyman–Pearson lemma, that the acceptance set $\mathcal{A}_l$ in Equation (1) is optimal in terms of having the smallest volume among all the $1 - \alpha$ acceptance sets for testing $H_0 : c = l$.

As for the usual confidence sets, it is desirable that, among the confidence sets $\mathcal{C}(\mathbf{y}_1), \mathcal{C}(\mathbf{y}_2), \ldots$ for the corresponding unknown true classes $c_1, c_2, \ldots \in C$ of the infinitely many future $\mathbf{y}_j$ with distribution $N(\mu_{c_j}, \Sigma_{c_j})$ $(j = 1, 2, \ldots)$, at least $1 - \alpha$ proportion will contain the true $c_j$'s. That is, it is desirable that

$$\liminf_{N\to\infty} \frac{1}{N} \sum_{j=1}^{N} I_{\left\{c_j \in \mathcal{C}(\mathbf{y}_j)\right\}} \ge 1 - \alpha, \tag{3}$$

where $I_A$ denotes the indicator function of set $A$ and so $\frac{1}{N}\sum_{j=1}^{N} I_{\left\{c_j \in \mathcal{C}(\mathbf{y}_j)\right\}}$ is the proportion among the $N$ confidence sets $\mathcal{C}(\mathbf{y}_j)$ that contains the true classes $c_j$. It is shown in the Appendix A that the property in Equation (3) holds with equality.

The interpretation of the property in Equation (3) is similar to that of a standard confidence set. The noteworthy difference is that the confidence sets $\mathcal{C}(\mathbf{y}_j)$ are for possibly different parameters $c_j$ $(j = 1, 2, \ldots)$. In addition, note that, for each $j$, $\mathcal{C}(\mathbf{y}_j)$ is a standard $1 - \alpha$ level confidence set for $c_j$, with $\mathbf{y}_j$ being the only source of randomness.

Figure 1 gives an illustrative example with $k = 3$, $p = 2$:

$$\mu_1 = \begin{pmatrix} 5.01 \\ 3.43 \end{pmatrix}, \quad \mu_2 = \begin{pmatrix} 5.94 \\ 2.77 \end{pmatrix}, \quad \mu_3 = \begin{pmatrix} 6.59 \\ 2.97 \end{pmatrix},$$

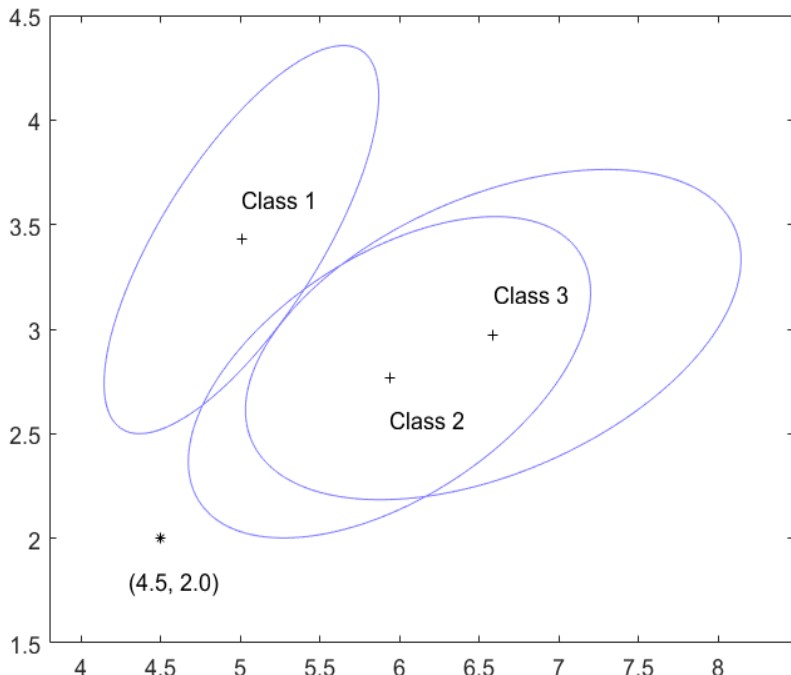

**Figure 1.** The acceptance sets for the three classes with known $\mu_i$ and $\Sigma_i$.

$$\Sigma_1 = \begin{pmatrix} 0.124, & 0.099 \\ 0.099, & 0.144 \end{pmatrix}, \ \Sigma_2 = \begin{pmatrix} 0.266, & 0.085 \\ 0.085, & 0.098 \end{pmatrix}, \ \Sigma_3 = \begin{pmatrix} 0.404, & 0.094 \\ 0.094, & 0.104 \end{pmatrix},$$

and $\alpha = 5\%$ (and so $\lambda = \chi^2_{2,0.95} = 5.991$). Specifically, the acceptance set $\mathcal{A}_l$ in Equation (1) is represented in Figure 1 by the ellipsoidal region centred at $\mu_l$, marked by '+', $l = 1, 2, 3$. If $\mathbf{y} \in \mathcal{A}_l$ then $l$ is an element of the confidence set $\mathcal{C}(\mathbf{y})$ given in Equation (1). Hence, the following four situations can occur. (a) $\mathbf{y}$ falls into only one $\mathcal{A}_l$ and so $\mathcal{C}(\mathbf{y})$ has a single class. For example, if $\mathbf{y} \in \mathcal{A}_1 \cap \mathcal{A}_2^c \cap \mathcal{A}_3^c$, then $\mathcal{C}(\mathbf{y}) = \{1\}$, i.e., $\mathbf{y}$ is classified as belonging to class 1. (b) $\mathbf{y}$ falls into two $\mathcal{A}_l$'s but not the other one, and so $\mathcal{C}(\mathbf{y})$ contains two classes. For example, if $\mathbf{y} \in \mathcal{A}_1 \cap \mathcal{A}_2 \cap \mathcal{A}_3^c$, then $\mathcal{C}(\mathbf{y}) = \{1, 2\}$, i.e., $\mathbf{y}$ is classified as belonging to possibly classes 1 or 2. (c) $\mathbf{y}$ falls into all the three $\mathcal{A}_l$'s, i.e., $\mathbf{y} \in \mathcal{A}_1 \cap \mathcal{A}_2 \cap \mathcal{A}_3$, and so $\mathbf{y}$ is classified as belonging to possibly all three classes. (d) $\mathbf{y}$ falls outside all the $\mathcal{A}_l$'s, i.e., $\mathbf{y} \in \mathcal{A}_1^c \cap \mathcal{A}_2^c \cap \mathcal{A}_3^c$, and so $\mathcal{C}(\mathbf{y}) = \varnothing$ and $\mathbf{y}$ is classified as not belonging to any one of the three classes. There is nothing wrong with this last classification since this $\mathbf{y}$ is judged not to be from the class $l$ by the acceptance set $\mathcal{A}_l$ for each $l$, though such a $\mathbf{y}$ must be rare in order to guarantee the property in Equation (3). On the other hand, since it is known that $\mathbf{y}$ is from one of the $k$ classes, it is sensible to classify $\mathbf{y}$ according to any reasonable classifier, e.g., a Bayesian classifier illustrated in the next paragraph. As the resultant confidence set $\mathcal{C}^*(\mathbf{y})$ from this augmentation contains $\mathcal{C}(\mathbf{y})$, the property in Equation (3) clearly still holds for $\mathcal{C}^*(\mathbf{y})$.

For example, if $\mathbf{y} = (4.5, 2.0)^T$, which is marked by '*' in Figure 1, then $\mathcal{C}(\mathbf{y}) = \varnothing$. The Bayesian classifier and the augmented confidence set $\mathcal{C}^*(\mathbf{y})$ can be worked out in the following way. Assume a non-informative prior $\pi(1) = \pi(2) = \pi(3) = 1/3$ about the class $c$ of $\mathbf{y}$, then the posterior probability of $\mathbf{y}$ belonging to class $l$ is given by

$$p(l|\mathbf{y}) = \pi(l)f(\mathbf{y}, l)/p(\mathbf{y}) = f(\mathbf{y}, l)/[3p(\mathbf{y})],$$

where $f(\mathbf{y}, l)$ is the probability density function of $N(\mu_l, \Sigma_l)$ and $p(\mathbf{y})$ is the marginal density of $\mathbf{y}$ and so does not depend on $l$. Hence, the Bayesian classifier classifies $\mathbf{y}$ to the class $i_0$ that satisfies

$f(\mathbf{y}, i_0) = \max_{l \in C} f(\mathbf{y}, l)$. For $\mathbf{y} = (4.5, 2.0)^T$, we have $f(\mathbf{y}, 1) = 0.009$, $f(\mathbf{y}, 2) = 0.612$ and $f(\mathbf{y}, 3) = 0.046$. Hence, the Bayesian classifier and the augmented confidence set $C^*(\mathbf{y})$ classify $\mathbf{y}$ to class 2.

In this particular example, $\mathcal{A}_1$ and $\mathcal{A}_3$ do not intersect as seen from Figure 1 and so any future $\mathbf{y}$ will not be classified to be in both classes 1 and 3. This reflects the fact that the distributions of the classes 1 and 3 are quite different/separated and so easy to distinguish. On the other hand, the distributions of the classes 2 and 3 are similar and so hard to distinguish. As a result, $\mathcal{A}_2$ and $\mathcal{A}_3$ have a large overlap and hence many future $\mathbf{y}$'s will be classified as belonging to both classes 2 and 3.

## 4. Unknown $\mu_i$ and $\Sigma_i$

### 4.1. Methodology

Now, we consider the more realistic situation where both the values of $\mu_i$ and $\Sigma_i$ are unknown and so need to be estimated from the training data set $\mathcal{T}$, independent of the future observations $\mathbf{y}_j$ ($j = 1, 2, \ldots$) whose classes $c_j$ are unknown and need to be inferred.

The training data set $\mathcal{T} = \{\mathbf{x}_{i1}, \ldots, \mathbf{x}_{in_i}; i = 1, \ldots, k\}$ can be used to estimate $\mu_i$ and $\Sigma_i$ in the usual way: $\hat{\mu}_i = \frac{1}{n_i} \sum_{m=1}^{n_i} \mathbf{x}_{im}$, $\hat{\Sigma}_i = \frac{1}{n_i - 1} \sum_{m=1}^{n_i} (\mathbf{x}_{im} - \hat{\mu}_i)(\mathbf{x}_{im} - \hat{\mu}_i)^T$, $i = 1, \ldots, k$. It is known [38] that $\hat{\mu}_i \sim N(\mu_i, \Sigma_i / n_i)$, $(n_i - 1)\hat{\Sigma}_i = \sum_{m=1}^{n_i - 1} \mathbf{z}_{im} \mathbf{z}_{im}^T$ with $\mathbf{z}_{i1}, \ldots, \mathbf{z}_{i(n_i - 1)}$ being i.i.d. $N(\mathbf{0}, \Sigma_i)$ random vectors independent of $\hat{\mu}_i$.

Mimicking the confidence set in Equation (2), we construct the confidence set for the class $c$ of $\mathbf{y}$ as:

$$\mathcal{C}_{\mathcal{T}}(\mathbf{y}) = \left\{ l \in C : (\mathbf{y} - \hat{\mu}_l)^T \hat{\Sigma}_l^{-1} (\mathbf{y} - \hat{\mu}_l) \leq \lambda \right\}, \tag{4}$$

where $\lambda$ is a suitably chosen critical constant whose determination is considered next.

As in Section 3, it is desirable that the proportion of the future confidence sets $\mathcal{C}_{\mathcal{T}}(\mathbf{y}_j)$ ($j = 1, 2, \ldots$) that include the true classes $c_j$ ($j = 1, 2, \ldots$) should be at least $1 - \alpha$:

$$\liminf_{N \to \infty} \frac{1}{N} \sum_{j=1}^{N} I_{\left\{ c_j \in \mathcal{C}_{\mathcal{T}}(\mathbf{y}_j) \right\}} \geq 1 - \alpha. \tag{5}$$

It is shown in the Appendix A that a sufficient condition for guaranteeing Inequality (5) is

$$\inf_{c_j \in C} E_{\mathbf{y}_j | \mathcal{T}} I_{\left\{ c_j \in \mathcal{C}_{\mathcal{T}}(\mathbf{y}_j) \right\}} \geq 1 - \alpha, \tag{6}$$

where $E_{\mathbf{y}_j | \mathcal{T}}$ denotes the conditional expectation with respect to the random variable $\mathbf{y}_j$ conditioning on the training data set $\mathcal{T}$ (or, equivalently, $\{(\hat{\mu}_1, \hat{\Sigma}_1), \ldots, (\hat{\mu}_k, \hat{\Sigma}_k)\}$).

Since the value of the expression on the left-hand side of the inequality in Inequality (6) depends on $\mathcal{T}$ and $\mathcal{T}$ is random, Inequality (6) cannot be guaranteed for each observed $\mathcal{T}$; more detailed explanation on this is given in the Appendix A. We therefore guarantee Inequality (6) with a large (close to 1) probability $\gamma$ with respect to the randomness in $\mathcal{T}$, which is shown in the Appendix A to be equivalent to

$$P_{\mathcal{T}} \left\{ \min_{1 \leq l \leq k} P_{\mathbf{w}_l | \mathbf{u}_l, \{\mathbf{v}_{lm}\}} \left\{ (\mathbf{w}_l - \mathbf{u}_l)^T \left( \frac{1}{n_l - 1} \sum_{m=1}^{n_l - 1} \mathbf{v}_{lm} \mathbf{v}_{lm}^T \right)^{-1} (\mathbf{w}_l - \mathbf{u}_l) \leq \lambda \right\} \geq 1 - \alpha \right\} \geq \gamma, \tag{7}$$

where

$$\mathbf{w}_l \sim N(\mathbf{0}, I_p), \quad \mathbf{u}_l \sim N(\mathbf{0}, I_p / n_l), \quad \mathbf{v}_{lm} \sim N(\mathbf{0}, I_p), \quad m = 1, \cdots, n_l - 1 \tag{8}$$

and all the $\mathbf{w}_l$'s, $\mathbf{u}_l$'s and $\mathbf{v}_{lm}$'s are independent. This in turn guarantees that

$$P_{\mathcal{T}} \left\{ \liminf_{N \to \infty} \frac{1}{N} \sum_{j=1}^{N} I_{\{c_j \in \mathcal{C}_{\mathcal{T}}(\mathbf{y}_j)\}} \geq 1 - \alpha \right\} \geq \gamma. \tag{9}$$

The interpretation of this statement is that, based on one observed training data set $\mathcal{T}$, one constructs confidence sets $\mathcal{C}_{\mathcal{T}}(\mathbf{y}_j)$ for the $c_j$'s of all future $\mathbf{y}_j$ ($j = 1, 2, \cdots$) and claims that at least $1 - \alpha$ proportion of these confidence sets do contain the true $c_j$'s. Then, we are $\gamma$ confident with respect to the randomness in the training data set $\mathcal{T}$ that the claim is correct.

It is noteworthy that for the classification problem considered in this paper a classifier is built from one training data set $\mathcal{T}$ and then used to classify a large number of future $\mathbf{y}_j$'s. Hence, the randomness in both the training data set $\mathcal{T}$ and the future $\mathbf{y}_j$'s need to be accounted for but in different ways. This is reflected in our approach by the two numbers $1 - \alpha$ and $\gamma$, analogous to the idea of [19,20] as pointed out in Section 2.

If we treat the two sources of randomness in $\mathbf{y}$ and $\mathcal{T}$ simultaneously on equal footing (instead of the approach given above), then it is straightforward to show that ([38], Section 5.2)

$$(\mathbf{y} - \hat{\boldsymbol{\mu}}_c)^T \hat{\Sigma}_c^{-1} (\mathbf{y} - \hat{\boldsymbol{\mu}}_c) \sim \frac{(n_c + 1)(n_c - 1)p}{n_c(n_c - p)} F_{p, n_c - p},$$

where $c$ is the true class of $\mathbf{y}$, and $F_{p, n_c - p}$ denotes an $F$ random variable with degrees of freedom $p$ and $n_c - p$. It follows therefore from Neyman's method that

$$\mathcal{C}(\mathcal{T}, \mathbf{y}) = \left\{ l \in C : (\mathbf{y} - \hat{\boldsymbol{\mu}}_l)^T \hat{\Sigma}_l^{-1} (\mathbf{y} - \hat{\boldsymbol{\mu}}_l) \leq \frac{(n_l + 1)(n_l - 1)p}{n_l(n_l - p)} f_{p, n_l - p, 1 - \alpha} \right\} \tag{10}$$

is a $1 - \alpha$ confidence set for $c$, where $f_{p, n_l - p, 1 - \alpha}$ is the $1 - \alpha$ quantile of $F_{p, n_l - p}$. However, this confidence set has the following coverage frequency interpretation. Collect one training data set $\mathcal{T}$ and the feature $\mathbf{y}$ of one future object, both of which are then used to compute the confidence set $\mathcal{C}(\mathcal{T}, \mathbf{y})$ for the class $c$ of $\mathbf{y}$; then, the frequency of the confidence sets that contain the true $c$'s is $1 - \alpha$ among a large number of confidence sets constructed in this way. Note that, in this construction, one training data set $\mathcal{T}$ is used only once with one future $\mathbf{y}$ to produce one confidence set $\mathcal{C}(\mathcal{T}, \mathbf{y})$, and so the randomness in one $\mathcal{T}$ and the randomness in one future $\mathbf{y}$ are treated on equal footing. This is clearly different from what is considered in this paper and how statistical classification is used in most applications: only one training data set $\mathcal{T}$ is used to construct a classifier, which is then used repeatedly in classification of a large number of future objects with observed $\mathbf{y}$ values. Hence, our proposed new method treats the two sources of randomness in $\mathcal{T}$ and future $\mathbf{y}$'s differently.

### 4.2. Algorithm for Computing $\lambda$

We now consider how to compute the critical constant $\lambda$ so that the probability $P_{\mathcal{T}}$ in Equation (7) is equal to $\gamma$. This is accomplished by simulation in the following way. From the distributions given in Equation (8), in the $s$th repeat of simulation, $s = 1, \ldots, S$, generate independent

$$\mathbf{u}_l^s \sim N(\mathbf{0}, I_p / n_l), \quad \mathbf{v}_{l1}^s, \ldots, \mathbf{v}_{l(n_l - 1)}^s \sim N(\mathbf{0}, I_p); \quad l = 1, \ldots, k$$

and find the $\lambda = \lambda_s$ so that

$$\min_{1 \leq l \leq k} P_{\mathbf{w}_l \mid \mathbf{u}_l^s, \{\mathbf{v}_{lm}^s\}} \left\{ (\mathbf{w}_l - \mathbf{u}_l^s)^T \left( \frac{1}{n_l - 1} \sum_{m=1}^{n_l - 1} \mathbf{v}_{lm}^s \mathbf{v}_{lm}^{s\,T} \right)^{-1} (\mathbf{w}_l - \mathbf{u}_l^s) \leq \lambda_s \right\} = 1 - \alpha. \tag{11}$$

Repeat this $S$ times to get $\lambda_1, \ldots, \lambda_S$ and order these as $\lambda_{[1]} \leq \ldots \leq \lambda_{[S]}$. It is well known [39] that $\lambda_{[\gamma S]}$ converges to the required critical constant $\lambda$ with probability one as $S \to \infty$. Hence, $\lambda_{[\gamma S]}$ is used as the required critical constant $\lambda$ for a large $S$ value, 10,000 say.

To find the $\lambda_s$ in Equation (11) for each $s$, we also use simulation in the following way. Generate independent random vectors $\{\mathbf{w}_{lq} : q = 1, \ldots, Q; l = 1, \ldots, k\}$ from $N(\mathbf{0}, I_p)$, where $Q$ is the number of simulations for finding $\lambda_s$. For each $l$, denote

$$t_{lq}^s = (\mathbf{w}_{lq} - \mathbf{u}_l^s)^T \left( \frac{1}{n_l - 1} \sum_{m=1}^{n_l - 1} \mathbf{v}_{lm}^s \mathbf{v}_{lm}^{s\ T} \right)^{-1} (\mathbf{w}_{lq} - \mathbf{u}_l^s), \quad q = 1 \ldots, Q$$

and their ordered values as $t_{l[1]}^s \leq \ldots \leq t_{l[Q]}^s$. Then, it is clear that $t_{l[(1-\alpha)Q]}^s$ is the sample $1 - \alpha$ quantile of $(\mathbf{w}_l - \mathbf{u}_l^s)^T \left( \frac{1}{n_l-1} \sum_{m=1}^{n_l-1} \mathbf{v}_{lm}^s \mathbf{v}_{lm}^{s\ T} \right)^{-1} (\mathbf{w}_l - \mathbf{u}_l^s)$ in which only $\mathbf{w}_l$ is random, and so $t_{l[(1-\alpha)Q]}^s$ converges to the population $1 - \alpha$ quantile $\lambda_{sl}$ with probability one as $Q \to \infty$, where $\lambda_{sl}$ satisfies

$$P_{\mathbf{w}_l \mid \mathbf{u}_l^s, \{\mathbf{v}_{lm}^s\}} \left\{ (\mathbf{w}_l - \mathbf{u}_l^s)^T \left( \frac{1}{n_l - 1} \sum_{m=1}^{n_l - 1} \mathbf{v}_{lm}^s \mathbf{v}_{lm}^{s\ T} \right)^{-1} (\mathbf{w}_l - \mathbf{u}_l^s) \leq \lambda_{sl} \right\} = 1 - \alpha.$$

Hence, $\max_{1 \leq l \leq k} t_{l[(1-\alpha)Q]}^s$ converges to $\max_{1 \leq l \leq k} \lambda_{sl} = \lambda_s$ as $Q \to \infty$ and is used as an approximation to $\lambda_s$ for a large $Q$ value, 10,000 say.

It is noteworthy that $\lambda$ depends only on $\gamma, \alpha, p, k, n_1, \ldots, n_k$ (and the numbers of simulations $S$ and $Q$ which determine the numerical accuracy of $\lambda$ due to simulation randomness). One can download from [40] our R computer program `ConfidenceSetClassifier.R` that implements this simulation method of computing the critical constant $\lambda$. While it is expected that larger values of $S$ and $Q$ will produce a more accurate $\lambda$ value, it must be pointed out that there is no easy way to assess how the accuracy of $\lambda$ depends on the values of $S$ and $Q$. One practical way is to compute several $\lambda$ values using different random seeds in the simulation for given $S$ and $Q$, which form a random sample from the population of possible $\lambda$ values. These $\lambda$ values provide information on the variability among the possible $\lambda$ values produced by the simulation method, and so accuracy of $\lambda$ due to simulation randomness. See more details in Section 5.

As in Section 3, the confidence set $\mathcal{C}_{\mathcal{T}}(\mathbf{y})$ in Equation (4) may be empty for a $\mathbf{y}$ and so $\mathbf{y}$ is classified as not belonging to any of the $c$ classes. As discussed in Section 3, there is nothing wrong with this, but it is sensible to classify such a $\mathbf{y}$ according to any reasonable classifier. The resultant confidence set $\mathcal{C}_{\mathcal{T}}^*(\mathbf{y})$ from this augmentation contains $\mathcal{C}_{\mathcal{T}}(\mathbf{y})$, and so Inequality (9) still holds for $\mathcal{C}_{\mathcal{T}}^*(\mathbf{y})$.

## 5. An Illustrative Example

The famous `iris` data set introduced by [41] is used in this section to illustrate the method proposed in this paper. The data set is simple but serves the purpose of illustration nevertheless. It contains $k = 3$ classes representing the three species/classes of Iris flowers (1 = setosa, 2 = versicolor, 3 = virginica), and has $n_i = 50$ observations from each class in $\mathcal{T}$. Each observation gives the measurements (in centimetres) of the four variables: sepal length and width, and petal length and width. The data set `iris` can be found in ([42], Chapter 10) for example, and is also in the R base package.

First, we assume that only the first two measurements, sepal length and width, are used for classification in order to easily illustrate the method since the acceptance sets $\mathcal{A}_l, l = 1, 2, 3$ are

two-dimensional and so can be easily plotted in this case. Based on the fifty observations on $p = 2$ measurements from each of the three classes, one can calculate that

$$\hat{\mu}_1 = \begin{pmatrix} 5.01 \\ 3.43 \end{pmatrix}, \ \hat{\mu}_2 = \begin{pmatrix} 5.94 \\ 2.77 \end{pmatrix}, \ \hat{\mu}_3 = \begin{pmatrix} 6.59 \\ 2.97 \end{pmatrix},$$

$$\hat{\Sigma}_1 = \begin{pmatrix} 0.124, & 0.099 \\ 0.099, & 0.144 \end{pmatrix}, \ \hat{\Sigma}_2 = \begin{pmatrix} 0.266, & 0.085 \\ 0.085, & 0.098 \end{pmatrix} \text{ and } \hat{\Sigma}_3 = \begin{pmatrix} 0.404, & 0.094 \\ 0.094, & 0.104 \end{pmatrix}.$$

In the example in Section 3, these are used as the known values of $\mu_i$ and $\Sigma_i$ for the three classes. For $\alpha = 5\%$ and $\gamma = 95\%$, the critical constant $\lambda$ in Equation (7) is computed by our R program to be 9.175 using $S = 10,000$ and $Q = 10,000$. The confidence set $\mathcal{C}_{\mathcal{T}}(\mathbf{y})$ in (4) is based on the acceptance sets $\mathcal{A}_l = \left\{ \mathbf{y} \in R^p : (\mathbf{y} - \hat{\mu}_l)^T \hat{\Sigma}_l^{-1} (\mathbf{y} - \hat{\mu}_l) \leq \lambda \right\}$, $l = 1, 2, 3$, which are plotted in Figure 2 by the ellipsoidal region centred at $\hat{\mu}_l$, marked by '+', $l = 1, 2, 3$. These ellipsoidal regions are larger than, but have the same centers and shapes as, the corresponding ellipsoidal regions given in Figure 1 of Section 3. This reflects the fact that the underlying multivariate normal distributions have been estimated from the training data $\mathcal{T}$ in this case and so involve uncertainty, while the distributions in Section 3 are assumed to be known.

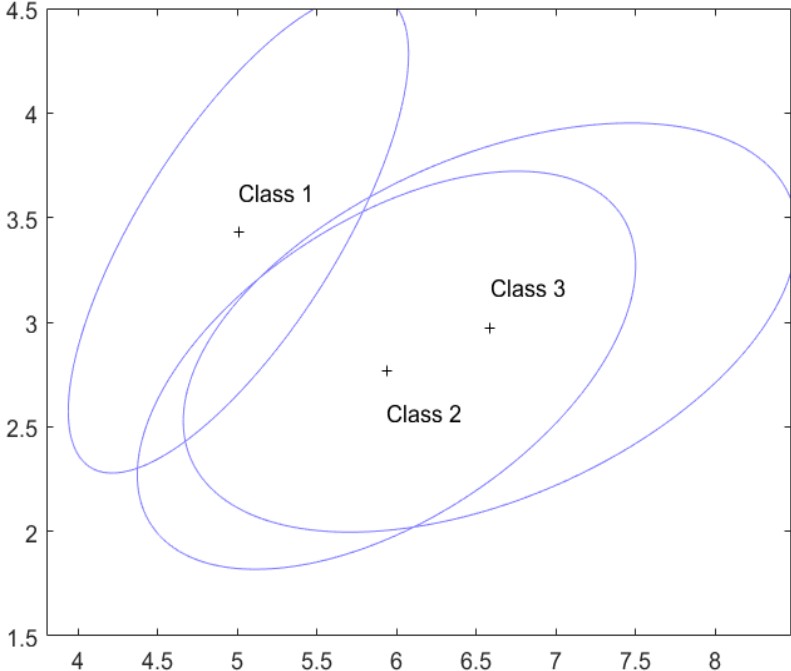

**Figure 2.** The acceptance sets for the three classes with estimated $\mu_i$ and $\Sigma_i$.

The index $l$ is an element of the confidence set $\mathcal{C}_{\mathcal{T}}(\mathbf{y})$ in Equation (3) if and only if $\mathbf{y} \in \mathcal{A}_l$. Hence, the following four situations can occur, similar to those in Section 3. (a) $\mathbf{y}$ falls into only one $\mathcal{A}_l$ and so $\mathcal{C}_{\mathcal{T}}(\mathbf{y})$ has only one class. (b) $\mathbf{y}$ falls into two $\mathcal{A}_l$'s but not the other one, and so $\mathcal{C}_{\mathcal{T}}(\mathbf{y})$ contains two classes. (c) $\mathbf{y}$ falls into all the three $\mathcal{A}_l$'s, i.e., $\mathbf{y} \in \mathcal{A}_1 \cap \mathcal{A}_2 \cap \mathcal{A}_3$, and so $\mathbf{y}$ is classified as belonging to possibly all three classes. (d) $\mathbf{y}$ falls outside all the $\mathcal{A}_l$'s, i.e., $\mathbf{y} \in \mathcal{A}_1^c \cap \mathcal{A}_2^c \cap \mathcal{A}_3^c$, and so $\mathcal{C}_{\mathcal{T}}(\mathbf{y}) = \varnothing$ and $\mathbf{y}$ is classified as not belonging to any one of the three classes.

From Figure 2, it is clear that $\mathcal{A}_1 \cap \mathcal{A}_2 \cap \mathcal{A}_3 \neq \varnothing$ and so for any future $\mathbf{y} \in \mathcal{A}_1 \cap \mathcal{A}_2 \cap \mathcal{A}_3$ the confidence set $\mathcal{C}_{\mathcal{T}}(\mathbf{y}) = \{1, 2, 3\}$ that is, $\mathbf{y}$ is judged to be possibly from any of the three classes.

As in Section 3, if $\mathbf{y}$ does not belong to any $\mathcal{A}_l, l = 1, 2, 3$, we compute the augmented confidence set $\mathcal{C}_{\mathcal{T}}^*(\mathbf{y})$ by using, for example, the naive Bayesian classifier with a non-informative prior that

classifies $\mathbf{y}$ to the class $i_0$ that satisfies $\hat{f}(\mathbf{y}, i_0) = \max_{l \in C} \hat{f}(\mathbf{y}, l)$, where $\hat{f}(\cdot, l)$ is the multivariate normal density function of the $l$th class with $\boldsymbol{\mu}_l$ and $\Sigma_l$ replaced by the estimates $\hat{\boldsymbol{\mu}}_l$ and $\hat{\Sigma}_l$, respectively.

To get some idea of how sensitive the critical constant $\lambda$ is to the simulation numbers $S$ and $Q$, we have computed $\lambda$ for various $(S, Q)$ with $\gamma = 0.95, \alpha = 0.05, p = 2, k = 3$ and $n_1 = n_2 = n_3 = 50$ on an ordinary Windows PC (Core (TM2) Due CPU P8400@2.26 GHz ). As it is expected that larger values of $S$ and $Q$ will produce more accurate $\lambda$ value, the results given in Table 1 indicate that the $\lambda$ value based on $(S, Q) = (10,000, \ 10,000)$, in comparison with the $\lambda$ value based on $(S, Q) = (20,000, \ 20,000)$, is accurate to at least the first decimal place and so probably sufficiently accurate for most real problems.

**Table 1.** Constant $\lambda$ and computation time $CT$ for various $(S, Q)$.

| $(S, Q)$ | (1000, 1000) | (10,000, 10,000) | (10,000, 20,000) | (10,000, 50,000) | (20,000, 10,000) | (20,000, 20,000) |
|---|---|---|---|---|---|---|
| $\lambda$ | 9.289 | 9.175 | 9.199 | 9.204 | 9.187 | 9.198 |
| $CT$ | 15 min | 25 h | 51 h | 139 h | 58 h | 102 h |

Alternatively, one can compute several $\lambda$ values for the given $S$ and $Q$ values using different random seeds to assess the accuracy of a $\lambda$ value computed. For example, fourteen $\lambda$ values based on $(S, Q) = (10,000, \ 10,000)$ based on fourteen different random seeds are computed to be: 9.231, 9.188, 9.172, 9.223, 9.192, 9.178, 9.203, 9.191, 9.198, 9.225, 9.182, 9.189, 9.224, 9.181, which form a sample of observations from the population distribution of all possible values of $\lambda$. This sample can then be used to infer the population and, in particular, the standard deviation of the population which gives the variability (or accuracy) of one $\lambda$ value from the population. The mean and standard deviation of this sample of fourteen observations are given by 9.198 and 0.0196, respectively, and so the $\lambda$ value based on $(S, Q) = (10,000, \ 10,000)$ is expected to be within the range $9.198 \pm 3 \times 0.0196$ using the "three-sigma" rule.

It is also worth emphasizing that only one $\lambda$ needs to be computed based on the observed training dataset $\mathcal{T}$ which is then used for classifications of all future objects. Hence, one can always increase $S$ and $Q$ to achieve better accuracy of $\lambda$ as required and computation time should not be of a great concern.

If all the four measurements are used in classification, then $p = 4$ and the acceptance sets $\mathcal{A}_l = \left\{ \mathbf{y} \in R^p : (\mathbf{y} - \hat{\boldsymbol{\mu}}_l)^T \hat{\Sigma}_l^{-1} (\mathbf{y} - \hat{\boldsymbol{\mu}}_l) \leq \lambda \right\}$, $l = 1, 2, 3$ are four dimensional ellipsoidal balls and so cannot be drawn. Nevertheless, the confidence set $\mathcal{C}_{\mathcal{T}}(\mathbf{y})$ in Equation (4) is still valid and can be computed easily for a given $\mathbf{y}$. For $\alpha = 5\%, \gamma = 95\%, p = 4, k = 3$ and $n_1 = n_2 = n_3 = 50$, the critical constant $\lambda$ in Equation (8) is computed by our R program to be $\lambda = 14.367$ using $S = 10,000$ and $Q = 10,000$. Now, suppose a future Iris flower has measurements $\mathbf{y} = (4.5, 3.5, 1.4, 0.27)$. Then, it is easy to check that $\mathbf{y} \in \mathcal{A}_1$ since $(\mathbf{y} - \hat{\boldsymbol{\mu}}_1)^T \hat{\Sigma}_1^{-1} (\mathbf{y} - \hat{\boldsymbol{\mu}}_1) = 5.915 \leq \lambda$, while $\mathbf{y} \notin \mathcal{A}_2 \cup \mathcal{A}_3$ since $(\mathbf{y} - \hat{\boldsymbol{\mu}}_l)^T \hat{\Sigma}_l^{-1} (\mathbf{y} - \hat{\boldsymbol{\mu}}_l) > \lambda$ for both $l = 2$ and 3. Hence, the confidence set $\mathcal{C}_{\mathcal{T}}(\mathbf{y})$ in (4) is $\{1\}$, that is, this Iris flower is classified as from class 1, i.e., Setosa.

## 6. A Simulation Study

In this section, a simulation study is carried out to illustrate the desirable feature of the confidence-set based classifier (CS) proposed in this paper, and to highlight its differences from the following popular classifiers: classification tree (CT, implemented using R package `tree`), multinomial logistic regression (MLR, implemented using R package `nnet`), support vector machine (SVM, implemented using R package `e1071`) and naive Bayes (NB, implemented using R package `e1071`). The setting $k = 3, p = 2, n_1 = n_2 = n_3 = 50, \gamma = 0.95$ and $\alpha = 0.05$ is considered following the illustrative example in the last section.

Three configurations of the $k = 3$ classes are considered in the simulation study. For the $\boldsymbol{\mu}_i$ and $\Sigma_i$ given in the example in Section 3, the first configuration (CONF1) has the $k = 3$ normal distributions given by $N(\boldsymbol{\mu}_1 + (0, -0.5)^T, \Sigma_1)$, $N(\boldsymbol{\mu}_2, \Sigma_2)$ and $N(\boldsymbol{\mu}_3, \Sigma_3)$. The second configuration

(CONF2) has the distributions $N(\boldsymbol{\mu}_1, \Sigma_1)$, $N(\boldsymbol{\mu}_2, \Sigma_2)$ and $N(\boldsymbol{\mu}_3, \Sigma_3)$. The third configuration (CONF3) has the distributions $N(\boldsymbol{\mu}_1, \Sigma_1)$, $N(\boldsymbol{\mu}_2, \Sigma_2)$ and $N(\boldsymbol{\mu}_3 + (1.0, 0.5)^T, \Sigma_3)$. CONF1 represents the situation that all the $k = 3$ classes are quite similar and thus hard to distinguish. CONF2 represents the situation that two of the $k = 3$ classes (i.e., classes 2 and 3) are quite similar but quite different from the other class (i.e., class 1). In CONF3, all the $k = 3$ classes are quite different and thus relatively easy to distinguish in comparison to CONF1 and CONF2.

For each configuration of the three population distributions, a random sample of size $n_i = 50$ is generated from each class/distribution to form the training data set $\mathcal{T}$ which is then used to train the classifiers CS, CT, MLR, SVM and NB. Each classifier is then used to classify $N = 3000$ future objects, with 1000 generated from each of the three classes/distributions; the proportion of correct classification, $\zeta$, of the $N = 3000$ objects is recorded. For CS, the average size $M$ of the confidence sets for the $N = 3000$ objects is also recorded; note that all the other classifiers classify each future object to only one class. This process is repeated for 100 times to produce $\zeta_1, \cdots, \zeta_{100}$ for each classifier, and $M_1, \cdots, M_{100}$ for CS only. Denote $\hat{\gamma} = \frac{1}{100} \sum_{i=1}^{100} I_{\{\zeta_i \geq 1-\alpha\}}$ and $\bar{\zeta} = \frac{1}{100} \sum_{i=1}^{100} \zeta_i$ for each classifier, and $\bar{M} = \frac{1}{100} \sum_{i=1}^{100} M_i$ for CS. The results on $\hat{\gamma}$, $\bar{\zeta}$ and $\bar{M}$ are given in Table 2, with the corresponding standard deviations given in brackets. One can download from [40] our R computer program `SimulationStudyF.R` that implements this simulation study.

**Table 2.** Simulation results. Abbreviations are defined in the text.

| | CS | | | CT | | MLR | | SVM | | NB | |
|---|---|---|---|---|---|---|---|---|---|---|---|
| | $\hat{\gamma}$ | $\bar{\zeta}$ | $\bar{M}$ | $\hat{\gamma}$ | $\bar{\zeta}$ | $\hat{\gamma}$ | $\bar{\zeta}$ | $\hat{\gamma}$ | $\bar{\zeta}$ | $\hat{\gamma}$ | $\bar{\zeta}$ |
| CONF1 | 1.00 | 0.98 | 2.06 | 0.00 | 0.71 | 0.00 | 0.77 | 0.00 | 0.76 | 0.00 | 0.74 |
| | | (0.0066) | (0.1075) | | (0.0258) | | (0.0099) | | (0.0121) | | (0.0110) |
| CONF2 | 1.00 | 0.98 | 1.67 | 0.00 | 0.74 | 0.00 | 0.80 | 0.00 | 0.80 | 0.00 | 0.79 |
| | | (0.0062) | (0.0552) | | (0.0254) | | (0.0085) | | (0.0090) | | (0.0095) |
| CONF3 | 1.00 | 0.98 | 1.31 | 0.00 | 0.90 | 0.10 | 0.94 | 0.07 | 0.94 | 0.01 | 0.94 |
| | | (0.0060) | (0.0716) | | (0.0159) | | (0.0052) | | (0.0058) | | (0.0068) |

Due to the property in Inequality (9) of CS, one expects that $\hat{\gamma} \geq \gamma = 0.95$ for CS. This is indeed the case for each of the three configurations from the results in Table 2. Note, however, that $\hat{\gamma}$ is either equal or close to zero for all the other classifiers. This is the advantage of CS, by construction, over the other classifiers. To guarantee the property in Inequality (9), the size of the confidence set may be larger than one as indicated by the $\bar{M}$ values in Table 2, while all the other classifiers select only one class for each future object. The average size of the confidence set depends on the configuration of the $k = 3$ classes. As expected, $\bar{M}$ tends to be smaller when the $k = 3$ classes are easier to distinguish, but larger when the $k = 3$ classes are harder to distinguish. For example, CONF3 has a considerably smaller $\bar{M}$ than CONF1.

As CS has the property in Inequality (9), it is not surprising that $\bar{\zeta}$ is likely to be larger than $1 - \alpha = 0.95$, which is born out by the results in Table 2. However, for the other classifiers, the value of $\bar{\zeta}$ depends on how different the $k = 3$ classes are; $\bar{\zeta}$ tends to be larger when the $k = 3$ classes are more different and thus easier to distinguish. For example, CONF3 has a larger $\bar{\zeta}$ than CONF1.

## 7. Conclusions

This paper considers how to deal with the classification problem using the novel confidence set approach by adapting the idea of [19,20] for inference about the predictor values of the observed response values in a standard linear regression model. Specifically, confidence sets $\mathcal{C}_{\mathcal{T}}(\mathbf{y}_j)$ for the true classes $c_j$ of infinitely many future objects $\mathbf{y}_j$ ($j = 1, 2, \ldots$), based on one training data set $\mathcal{T}$, have been constructed so that, with confidence level $\gamma$ about the randomness in $\mathcal{T}$, the proportion of the $\mathcal{C}_{\mathcal{T}}(\mathbf{y}_j)$'s that contain the true $c_j$'s is at least $1 - \alpha$.

The intuitive motivation underlying this method is that, when an object is judged to be possibly from several classes, we should accept this objectively rather than forcing ourselves to pick just one

class, which entails a large chance of misclassification. By allowing an object to be classified as possibly from more than one class, the proportion of correct classification can be guaranteed to be at least $1 - \alpha$ with a large probability $\gamma$ about the randomness in the training data set $\mathcal{T}$. This 'guaranteed probability $\gamma$ about the randomness in $\mathcal{T}$' should be intuitive too since a $\mathcal{T}$ that is very misleading about the $k$ classes will likely produce a classifier that makes many wrong classifications, and so only $\gamma$ proportion of well behaved $\mathcal{T}$ will produce a classifier that give at least $1 - \alpha$ future correct classifications.

The two sources of randomness, those in the training data $\mathcal{T}$ and in future objects $\mathbf{y}_j$, have been treated differently to reflect the fact that a classifier is built from one training data set $\mathcal{T}$ and then used to classify many future objects $\mathbf{y}_j$. If the two sources of randomness are treated on equal footing, then the confidence set in Equation (10) should be used, which has a very different coverage frequency interpretation.

In this paper, the objects $\mathbf{y}$ from each class are assumed to follow a multivariate normal distribution. How the proposed method can be generalized to, or may be affected by, non-normal distributions, such as the elliptically contoured distribution [38] (p. 47) is interesting and warrants further research.

A frequentist approach is proposed in this paper. One wonders whether a corresponding Bayesian approach is easier to construct. In a Bayesian approach, one uses the posterior distribution $\pi(c_j \mid \mathbf{y}_j, \mathcal{T})$ to make an inference about the true class $c_j$ of the future object $\mathbf{y}_j$. In particular, one can easily construct a Bayesian credible set $\mathcal{C}_B(\mathbf{y}_j)$ for $c_j$ such that $P\{c_j \in \mathcal{C}_B(\mathbf{y}_j) \mid \mathbf{y}_j, \mathcal{T}\} \geq 1 - \alpha$. However, it is not at all clear whether this construction guarantees that

$$\liminf_{N \to \infty} \frac{1}{N} \sum_{j=1}^{N} I_{\left\{c_j \in \mathcal{C}_B(\mathbf{y}_j)\right\}} \geq 1 - \alpha$$

since it can be shown that $\pi(c_i, c_j \mid \mathbf{y}_i, \mathbf{y}_j, \mathcal{T}) \neq \pi(c_i \mid \mathbf{y}_i, \mathcal{T})\pi(c_j \mid \mathbf{y}_j, \mathcal{T})$, i.e., the posterior distributions of $c_i$ and $c_j$ for the two future objects $\mathbf{y}_i$ and $\mathbf{y}_j$ are not independent. Nevertheless, Bayesian approach warrants further research.

**Author Contributions:** W.L., F.B., N.S., J.P., and A.J.H. all contributed to the writing of the paper, the data analysis, and the implementation of the simulations study.

**Acknowledgments:** We would like to thank the referees for critical and constructive comments.

**Funding:** The authors declare no funding.

**Conflicts of Interest:** The authors declare no conflict of interest.

**Appendix A. Mathematical Details**

In this appendix, we first show that the property in Equation (3) holds with equality. Note that we have

$$
\begin{aligned}
& \lim_{N \to \infty} \frac{1}{N} \sum_{j=1}^{N} I_{\left\{c_j \in \mathcal{C}(\mathbf{y}_j)\right\}} \\
=~ & \lim_{N \to \infty} \frac{1}{N} \sum_{j=1}^{N} P\left\{c_j \in \mathcal{C}(\mathbf{y}_j)\right\} \\
=~ & \lim_{N \to \infty} \frac{1}{N} \sum_{j=1}^{N} P\left\{(\mathbf{y}_j - \boldsymbol{\mu}_{c_j})^T \Sigma_{c_j}^{-1} (\mathbf{y}_j - \boldsymbol{\mu}_{c_j}) \leq \lambda\right\} \\
=~ & \lim_{N \to \infty} \frac{1}{N} \sum_{j=1}^{N} (1 - \alpha) = 1 - \alpha,
\end{aligned}
$$

where the first equality above follows from the classical strong law of large numbers ([43], p. 333), the second from the definition of $\mathcal{C}(\mathbf{y}_j)$ in Equation (2), and the third from $\lambda = \chi^2_{p,1-\alpha}$. This completes the proof.

Next, we show that

$$\inf_{c_j \in C} E_{\mathbf{y}_j|\mathcal{T}} I_{\{c_j \in \mathcal{C}_\mathcal{T}(\mathbf{y}_j)\}} \geq 1 - \alpha \text{ implies } \liminf_{N \to \infty} \frac{1}{N} \sum_{j=1}^{N} I_{\{c_j \in \mathcal{C}_\mathcal{T}(\mathbf{y}_j)\}} \geq 1 - \alpha, \tag{A1}$$

where $E_{\mathbf{y}_j|\mathcal{T}}$ denotes the conditional expectation with respect to the random variable $\mathbf{y}_j$ conditioning on the training data set $\mathcal{T}$ (or, equivalently, all the $\hat{\mu}_i$ and $\hat{\Sigma}_i$). We have from the classical strong law of large numbers [43] that

$$\lim_{N \to \infty} \frac{1}{N} \sum_{j=1}^{N} \left[ I_{\{c_j \in \mathcal{C}_\mathcal{T}(\mathbf{y}_j)\}} - E_{\mathbf{y}_j|\mathcal{T}} I_{\{c_j \in \mathcal{C}_\mathcal{T}(\mathbf{y}_j)\}} \right] = 0,$$

in which the conditional expectation $E_{\mathbf{y}_j|\mathcal{T}}$ is used since all the confidence sets $\mathcal{C}_\mathcal{T}(\mathbf{y}_j)$ ($j = 1, 2, \ldots$) use the same training data set $\mathcal{T}$. Hence,

$$\liminf_{N \to \infty} \frac{1}{N} \sum_{j=1}^{N} I_{\{c_j \in \mathcal{C}_\mathcal{T}(\mathbf{y}_j)\}}$$

$$= \lim_{N \to \infty} \frac{1}{N} \sum_{j=1}^{N} \left[ I_{\{c_j \in \mathcal{C}_\mathcal{T}(\mathbf{y}_j)\}} - E_{\mathbf{y}_j|\mathcal{T}} I_{\{c_j \in \mathcal{C}_\mathcal{T}(\mathbf{y}_j)\}} \right] + \liminf_{N \to \infty} \frac{1}{N} \sum_{j=1}^{N} E_{\mathbf{y}_j|\mathcal{T}} I_{\{c_j \in \mathcal{C}_\mathcal{T}(\mathbf{y}_j)\}}$$

$$= \liminf_{N \to \infty} \frac{1}{N} \sum_{j=1}^{N} E_{\mathbf{y}_j|\mathcal{T}} I_{\{c_j \in \mathcal{C}_\mathcal{T}(\mathbf{y}_j)\}}.$$

The required result in Expression (12) now follows immediately from

$$\frac{1}{N} \sum_{j=1}^{N} E_{\mathbf{y}_j|\mathcal{T}} I_{\{c_j \in \mathcal{C}_\mathcal{T}(\mathbf{y}_j)\}} \geq \inf_{c_j \in C} E_{\mathbf{y}_j|\mathcal{T}} I_{\{c_j \in \mathcal{C}_\mathcal{T}(\mathbf{y}_j)\}}$$

for any $N \geq 1$ since it is known that all the $c_j$'s are in $C$. This completes the proof.

Next, we provide a more tractable expression for $\inf_{c_j \in C} E_{\mathbf{y}_j|\mathcal{T}} I_{\{c_j \in \mathcal{C}_\mathcal{T}(\mathbf{y}_j)\}}$ in order to understand why Inequality (6) cannot be guaranteed for each observed $\mathcal{T}$. From the definition of $\mathcal{C}_\mathcal{T}(\mathbf{y})$ in Equation (4), we have

$$\inf_{c_j \in C} E_{\mathbf{y}_j|\mathcal{T}} I_{\{c_j \in \mathcal{C}_\mathcal{T}(\mathbf{y}_j)\}}$$

$$= \inf_{c_j \in C} P_{\mathbf{y}_j|\mathcal{T}} \{c_j \in \mathcal{C}_\mathcal{T}(\mathbf{y}_j)\}$$

$$= \inf_{c_j \in C} P_{\mathbf{y}_j|\mathcal{T}} \left\{ (\mathbf{y}_j - \hat{\mu}_{c_j})^T \hat{\Sigma}_{c_j}^{-1} (\mathbf{y}_j - \hat{\mu}_{c_j}) \leq \lambda \right\}$$

$$= \min_{1 \leq l \leq k} P_{\mathbf{w}_l | \mathbf{u}_l, \{\mathbf{v}_{lm}\}} \left\{ (\mathbf{w}_l - \mathbf{u}_l)^T \left( \frac{1}{n_l - 1} \sum_{m=1}^{n_l - 1} \mathbf{v}_{lm} \mathbf{v}_{lm}^T \right)^{-1} (\mathbf{w}_l - \mathbf{u}_l) \leq \lambda \right\}, \tag{A2}$$

where

$$\mathbf{w}_l = \Sigma_l^{-1/2} (\mathbf{y}_l - \mu_{c_l}) \sim N(\mathbf{0}, I_p), \tag{A3}$$

$$\mathbf{u}_l = \Sigma_l^{-1/2} (\hat{\mu}_l - \mu_{c_l}) \sim N(\mathbf{0}, I_p/n_l), \tag{A4}$$

$$\mathbf{v}_{lm} = \Sigma_l^{-1/2} \mathbf{z}_{lm} \sim N(\mathbf{0}, I_p), \ m = 1, \cdots, n_l - 1, \tag{A5}$$

with all the $\mathbf{w}_l$'s, $\mathbf{u}_l$'s and $\mathbf{v}_{lm}$'s being independent. Note that $\mathbf{w}_l$ depends on the future observation $\mathbf{y}_l$ but not the training data set $\mathcal{T}$, while $\mathbf{u}_l$ and $\{\mathbf{v}_{lm}\}$ depend on the training data set $\mathcal{T}$ but not the future observations.

Since the conditional probability in Equation (13) depends on the training data set $\mathcal{T}$ (via the random vectors $\mathbf{u}_l$ and $\{\mathbf{v}_{lm}\}$), Inequality (6), for any given value of $\lambda$, cannot be guaranteed for each observed training data set $\mathcal{T}$, i.e., $\mathbf{u}_l$ and $\{\mathbf{v}_{lm}\}$. For example, if the values of $\mathcal{T}$ are such that

$$(\mathbf{w}_l - \mathbf{u}_l)^T \left( \frac{1}{n_l - 1} \sum_{m=1}^{n_l-1} \mathbf{v}_{lm}\mathbf{v}_{lm}^T \right)^{-1} (\mathbf{w}_l - \mathbf{u}_l)$$

is substantially larger than $\lambda$ (for a given constant $\lambda$) for most possible values of $\mathbf{w}_l \sim N(\mathbf{0}, I_p)$, then the conditional probability in Equation (13) is smaller than $1/2$ and hence $1 - \alpha \in (1/2, 1)$.

We therefore guarantee Inequality (6) with a large (close to 1) probability $\gamma$ with respect to the randomness in $\mathcal{T}$, which is, from Equation (13), clearly equivalent to

$$P_{\mathcal{T}} \left\{ \min_{1 \le l \le k} P_{\mathbf{w}_l \mid \mathbf{u}_l, \{\mathbf{v}_{lm}\}} \left\{ (\mathbf{w}_l - \mathbf{u}_l)^T \left( \frac{1}{n_l - 1} \sum_{m=1}^{n_l-1} \mathbf{v}_{lm}\mathbf{v}_{lm}^T \right)^{-1} (\mathbf{w}_l - \mathbf{u}_l) \le \lambda \right\} \ge 1 - \alpha \right\} \ge \gamma, \quad \text{(A6)}$$

where $\mathbf{w}_l$, $\mathbf{u}_l$ and $\mathbf{v}_{lm}$ are given in Equations (14)–(16).

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
