# Peer review of "Confidence Sets for Statistical Classification"

_stats, doi:10.3390/stats2030024_

Round 1

Reviewer 1 Report

The addition of simulation results has improved the paper a lot.

Author Response

Thank you very much

Reviewer 2 Report

Still the answer  about parameters S and Q appears to be very experimental in nature.

Author Response

At the end of the 2nd paragraph on Page 17 of this revision, we have added

{\bf It must be pointed out that there is no easy way to assess how theaccuracy of $\lambda$ depends on the values of $S$ and $Q$.One practical way is to compute several $\lambda$-values usingdifferent random seeds in the simulation for given $S$ and $Q$.These $\lambda$-values provide information on the variability andso accuracy of $\lambda$ due to simulation randomness. See moredetails in Section 5} to pint out the fact there is no simple way toassess how the accuracy of $\lambda$ depends on the values of $S$ and $Q$.

Reviewer 3 Report

This manuscript discusses the classification problem and proposes a new classifier that classifies an observation into one or more categories based on a confidence set of the true class of the observation. The confidence set is constructed by inverting a family of acceptance sets. The proposed approach is innovative and interesting, especially useful when there is a similar chance of an observation belonging to two or more classes. I only have a minor comment: the use of ‘p’ is a bit confusing. On Page 22, ‘p’ is defined as ‘the proportion of correct classification’ while earlier ‘p’ is defined to be the dimension of the vector x_i. It would be clearer to use a different notation.

Author Response

In this revision, this $p$ has been replaced by $\zeta$.